# The phonon-modulated Jahn–Teller distortion of the nitrogen vacancy center in diamond

William P. Carbery [1] ✉, Camille A. Farfan[1], Ronald Ulbricht [2] ✉ & Daniel B. Turner [1] ✉

The negatively charged nitrogen vacancy (NV) center in diamond is an optically accessible material defect with a unique combination of spin and optical properties that has attracted interest in quantum-information sciences and as a design candidate for nanoscale quantum sensors. Here, we present time-resolved nonlinear optical spectroscopy measurements, conducted with ultrabroadband laser pulses, that reveal strong modulation of the excited-state by the longitudinal optical (LO) phonon of the diamond lattice. The LO phonon and its overtones geometrically distort neighboring NV centers, driving long lived (3.5 ps) excited state relaxation of coupled NV centers after the initial excitation and ultrafast (<150 fs) decay of the Jahn–Teller distortion. These observations elevate the LO phonon to an important tuning mode of the Jahn–Teller conical intersection and help resolve previous spectroscopy experiments that noted longer-lived excited-state dynamics.

The negatively charged nitrogen vacancy (NV) center in diamond is an optically accessible material defect with a long-lived spin coherence at room temperature. Its unique combination of spin and optical properties has attracted interest as a candidate for quantum-computer qubits[1] and nanoscale electromagnetic field sensors[2,3]. Next-generation device applications aside, the NV center attracts fundamental research interest as a platform for investigating coupled spin, vibration, and electronic interactions. These interactions are strongly dependent on the Jahn–Teller (JT) distortion of the NV center's excited state, which has been invoked to explain the spin-selective relaxation channels that ultimately establish the NV center's spin-polarization cycle[4–6]. The dynamic JT distortion also explains many of the NV center's anomalous optical properties. These include a slight spectral asymmetry between absorption and photoluminescence (PL) spectra[7–9], the non-unity polarization fidelity of the PL even at cryogenic temperatures[10,11] and the femtosecond timescale vibrational relaxation of the electronic excited state[12].

The geometric changes induced by the JT distortion form a conical intersection[13] composed of an $E \otimes e$ coupling between the orbitally degenerate $^3E$ excited state and an $e$-type vibrational mode that reduces the $C_{3v}$ symmetry of the NV center into three, inter-converting, equipotential wells with $C_{1h}$ symmetry[14,15]. The NV center's position within a rigid diamond lattice distinguishes the system from molecular JT distortions, as quasi-localized vibrational modes (qLVM) derived from the phonon continuum of the diamond lattice constitute the vibrational environment of the conical intersection.

Theoretical and experimental studies of the NV center suggest that these qLVMs mediate certain spin-selective intersystem crossing pathways that establish spin polarization in the spin-triplet electronic ground state $^3A_2$[4–6], thereby enabling optically-detected magnetic resonance experiments at convenient laboratory temperatures and timescales[16]. While ab initio simulations predict more than a dozen of these vibronic modes with frequencies ranging from 7 THz (30 meV) to 40 THz (165 meV), none have been directly measured in conventional vibrational spectroscopy measurements[17–19]. In addition, these computational models predict relatively weak vibronic coupling for the qLVMs, a prediction seemingly at odds with the strong nonadiabatic coupling and ultrafast dynamics associated with conical intersections.

This discrepancy is highlighted by a recent transient absorption (TA) spectroscopy experiment conducted by one of us that

[1]Department of Chemistry, New York University, New York, NY 10003, USA. [2]Max Planck Institute for Polymer Research, Ackermannweg 10, 55128 Mainz, Germany. ✉e-mail: wpcarbery@gmail.com; ulbricht@mpip-mainz.mpg.de; danielturner926@boisestate.edu

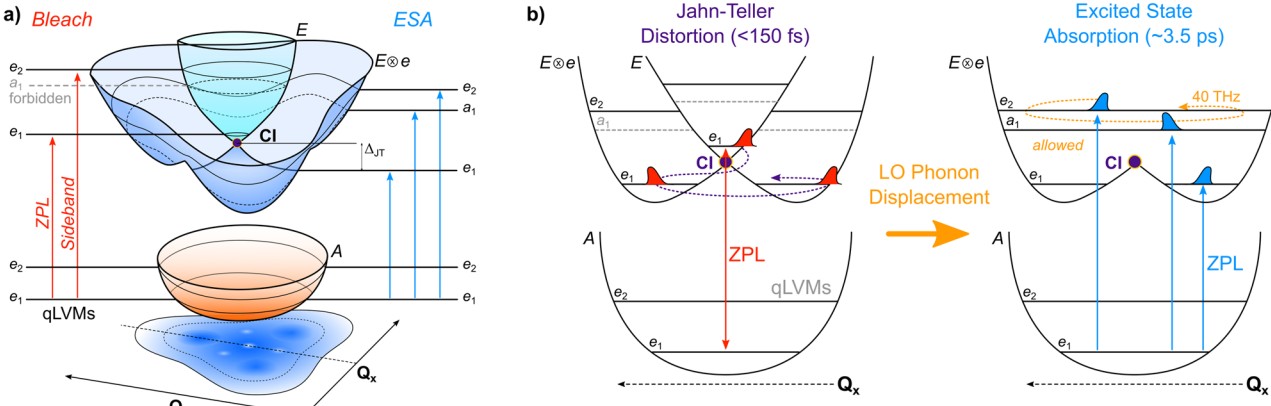

**Fig. 1 | Schematic of the adiabatic potential energy surfaces of the NV center. a** Three-dimensional representation of the ground state (*A*), excited state (*E*), and JT-distorted excited state (*E* ⊗ *e*). Adapted with permission from Abtew et al. and copyrighted by the American Physical Society[33]. Predicted qLVM levels ($e_1$, $e_2$, $a_1$) are highlighted in both the undistorted and distorted excited states, but are separated from their exact energetic positions for clarity. **b** Proposed surfaces illustrating the initial <150 fs dynamics of the NV center and the subsequent LO-phonon mediated displacement that activates neighboring NV centers and their associated ESA features.

demonstrated sub-50 fs electronic dynamics and vibrational damping occurring within the orbital manifold of the $^3E$ electronic state[12]. A wavepacket with those characteristics would pass through the JT conical intersection in less than half a vibrational cycle of the highest-frequency qLVM predicted by theory. Further complications arise from a 2013 report by Huxter et al., who measured time-resolved spectra of the NV center and inferred a vibrational relaxation time of 4.2 ps from coherent oscillations around the zero phonon line (ZPL)[20]. Both of the observed relaxation times are ill-suited to the description of a conical intersection environment derived from weakly coupled qLVMs, and furthermore exist on opposite extremes of typical relaxation rates observed for coherent vibrational wavepackets modulating molecular systems[21].

The NV center is a lattice system not a molecular system, and JT distortions trapped in solid-state materials are less researched relative to their molecular counterparts. In this contribution, we use ultrabroadband TA and two-dimensional electronic spectroscopy (2D ES) to investigate the coherent vibrational wavepackets that evolve on the coupled electronic states of the NV center. We have identified excited-state absorption (ESA) features that suggest LO-phonon activation of neighboring NV centers following optical excitation. These ESA features relax over the course of several picoseconds—thus confirming earlier reports of a picosecond "decay"—but evolve independently from the ground-state bleach (GSB) and delayed stimulated emission (SE) signal—thus confirming the sub-50 fs nonradiative decay of the excited state through the JT conical intersection. We find evidence of the computationally predicted qLVMs in the peak separations of individual 2D ES spectra but determine through coherence spectra that only the LO phonon and its overtones directly modulate the JT distortion.

A schematic representation of the NV center's relevant electronic states is shown in Fig. 1, alongside the adiabatic potential energy surfaces proposed for the ESA features observed in our spectra. Our findings suggest that the longitudinal optical (LO) phonon of the diamond lattice dominates the dynamics of the JT distortion at room temperature, bypassing the qLVMs and coupling neighboring NV centers through the diamond lattice. While the results of this report are unique to the NV center, the phonon modulation of JT distortions likely occurs in other lattice, solid-state, and topological materials.

## Results and discussion
The steady-state absorption and PL spectra ($\lambda_{exc}$ = 455 nm) of the NV center are presented in Fig. 2. The steady-state spectra show the diagnostic ZPL at 470.3 THz (637.4 nm) and broad phonon sidebands

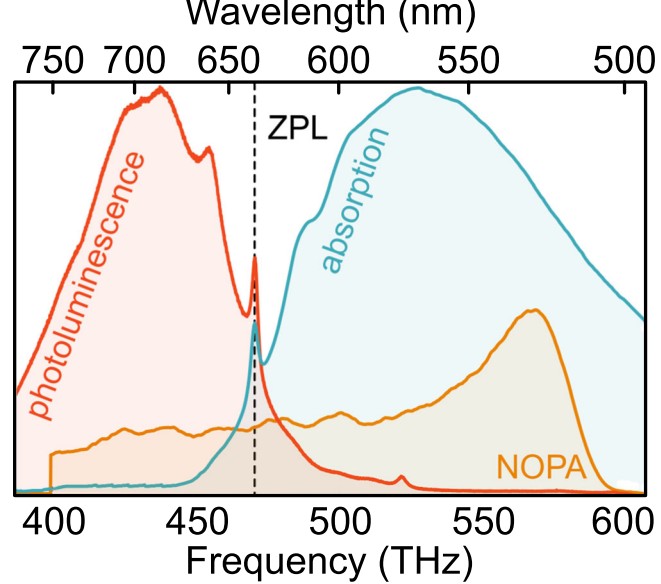

**Fig. 2 | Room-temperature absorption (blue), PL (red), and laser spectrum (orange).** The dotted line indicates the zero-phonon line (ZPL) excitation at 470.3 THz (637.4 nm).

in both absorption and emission[9]. Calibration of the measured absorption coefficient with the known NV center cross section[22] yields a defect concentration of about 1 part-per-million. Supplementary Information contains steady-state Raman spectra of the specific NV sample used in this report, alongside the Raman spectrum of a diamond blank for comparison.

In order to investigate the ultrafast dynamics of the NV center, we conducted a series of TA measurements using an ultrabroadband, 400–600 THz (500–750 nm) laser pulse. The large spectral bandwidth allows us to track the electronic and vibrational evolution of the NV center's excited state with sub-7 fs resolution. Coherent oscillations will arise in femtosecond TA spectra from vibrational wavepackets on both the ground and excited electronic states unless the bandwidth of the pump pulse exceeds the absorption linewidth[23–25]. Our bandwidth is sufficiently broad to suppress ground-state vibrational wavepackets, thus assuring that any observed vibrational modulation of the TA spectra stems exclusively from the excited-state dynamics.

**(a)**

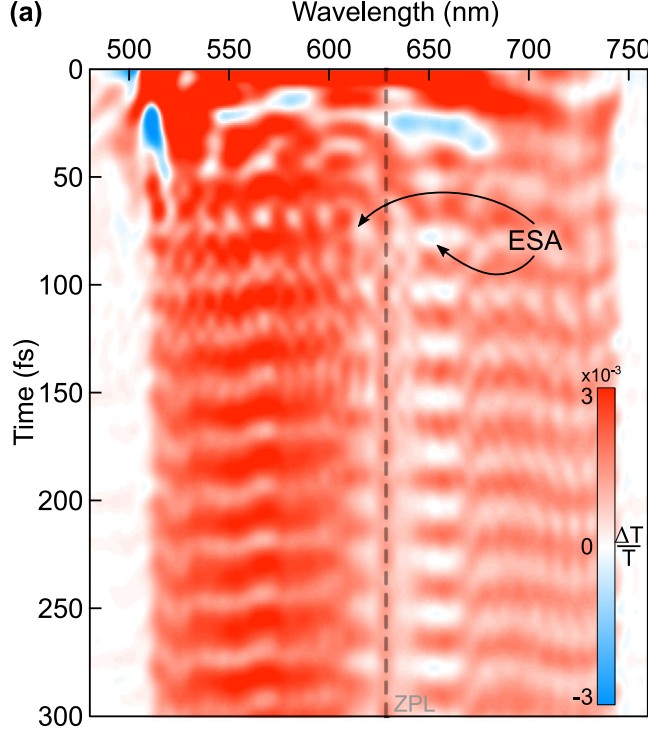

**(b)**

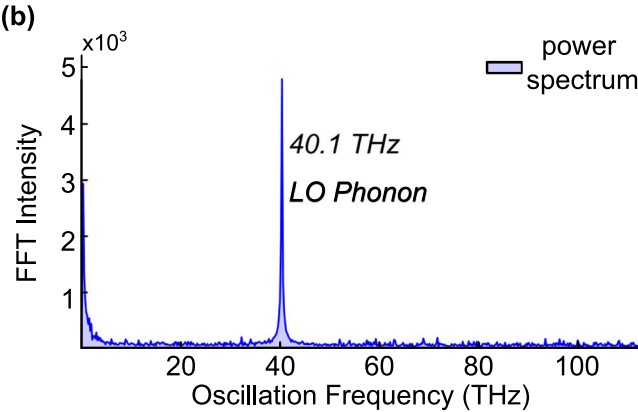

**Fig. 3 | TA data from 0 to 300 fs. a** The TA spectrum shows clear vibronic modulation and weak ESA features split by approximately 13 THz. **b** 1D power spectrum indicating a single vibrational mode at 40.1 THz (166 meV) attributed to the LO phonon.

The TA spectrum presented in Fig. 3, which we display as spectrally resolved differential transmission $\Delta T/T$, shows the dynamics of the NV center in the first 300 fs after excitation. The spectrum reveals strong oscillatory motion and a weak ESA feature on each side of the ZPL. Unlike GSB and SE signals—which are broadly comparable, respectively, to absorption and PL—ESA indicates absorptive transitions occurring after excitation.

The strong temporal modulation of the TA signal in Fig. 3 results from significant wavepacket motion. Fourier transformation of the oscillatory component of the TA spectrum, followed by summation over all wavelengths, produces a 1D power spectrum that reveals a solitary peak at 40.1 THz, attributable to the LO phonon of the NV center diamond lattice. In fact, the combined power spectra taken over multiple scanning ranges across three generations of our femtosecond spectrometer show only this single peak at 40.1 THz.

None of the peaks identified in theoretical work as qLVMs appear in the 1D power spectrum. This contrasts with the observation by Huxter et al., who identified more than ten oscillations in the frequency

range of the qLVMs[20]. Ultrashort laser pulses can excite coherent phonons with similar TA signatures via impulsively stimulated Raman scattering[26], and these types of measurements have previously uncovered the LO phonon in undoped diamond samples[27]. Using identical measurement conditions, we collected a TA spectrum of a blank diamond sample, and found that the intensity of the LO phonon peak in the NV center's 1D power spectrum is more than 4.5× greater than the intensity of the peak in a blank diamond sample of similar thickness.

The observation of LO phonon modulated ESA peaks in the TA spectra is unique to our ultrabroadband data and implies a connection to the JT conical intersection of the NV center. Ultrabroadband laser pulses are required to launch excited-state vibrational wavepackets, however, such pulses also prohibit the excitation specificity of narrowband TA spectroscopy. In order to resolve the ESA features in the ZPL region—and to investigate earlier findings of excitation-dependent delayed SE in the TA spectra[12,28]—we conducted 2D ES measurements of the NV center. Each 2D spectrum can be separated into regions, with GSB peaks in the upper right-hand corner, SE peaks in the upper left-hand corner, a series of narrow peaks comprising the ZPL region at an emission frequency of 470.3 THz, and ESA peaks along the diagonal at the ZPL cross peak. There is also an ESA peak at 575 THz originating from a triplet state at higher energies[29,30]. Identifying the ESA peaks requires 'phasing' of the spectra[31], a challenging procedure not conducted in previously reported 2D spectroscopy measurements of the NV center[20]. Phasing the 2D ES is particularly important for the NV center because the ESA peaks are symmetric across the ZPL and therefore disappear in magnitude spectra. Measurements of the ZPL linewidth and relaxation decay provide the foundation for discussing NV center dynamics, and the ESA features have the potential to disrupt interpretations of both measurements if not properly phased and analyzed.

As seen in the 9 ps spectrum presented in Fig. 4, 2D spectra can separate the GSB and SE peaks from the ESA signal. The four ESA peaks of interest are displaced from each other by 13.01 THz in the probe dimension and 17.58 THz in the excitation dimension, forming a rectangle surrounding the main ZPL peak at 470.3 THz. These peak separations relay information about the vibronic sublevels of the NV center's optical transitions because excitations to higher-lying vibrational modes are resolved as cross peaks when an ultrafast laser pulse encompasses the entire excitation and emission spectrum of the sample[32]. The ZPL of the NV center is convenient because it demarcates a transition with zero vibrational energy ($\nu_0$), allowing for straightforward assignment of the NV center vibronic levels by measuring the frequency difference of peaks relative to the ZPL. The full table of vibronic energy levels found in this way is presented in the Supplementary Information, while a smaller selection for comparison is presented in Table 1.

Of a total of eight vibrational frequencies found during this vibronic analysis, five of them correspond to qLVMs predicted by ab initio calculations performed on the NV center[33], while three others at 11.5, 16.0, and 20.6 THz match those found in published molecular dynamics simulations of the TA spectrum[34]. Two of the ESA peaks are red-shifted from the ZPL by 8.8 THz, a frequency that ab initio computations[33] predict to be the lowest-energy qLVM $a_1$, i.e. the JT tunneling splitting. Although the transition to the $a_1$ qLVM is forbidden in the initial excitation, symmetry breaking by the JT distortion allows the transition. This suggests that the ESA peaks arise from ZPL transitions between the ground ($^3A_2$) and excited state ($^3E$) after the JT distortion displaces the local geometry of the NV center. Depending on the simulation, the JT stabilization energy is below (25 meV)[33] or slightly above (42 meV)[5] the $a_1$ qLVM, and it would be interesting if future theoretical work could use the observed ESA features to predict optical signatures in 2D ES or related methods that would distinguish between these two scenarios. Nonadiabatic wavepacket propagation

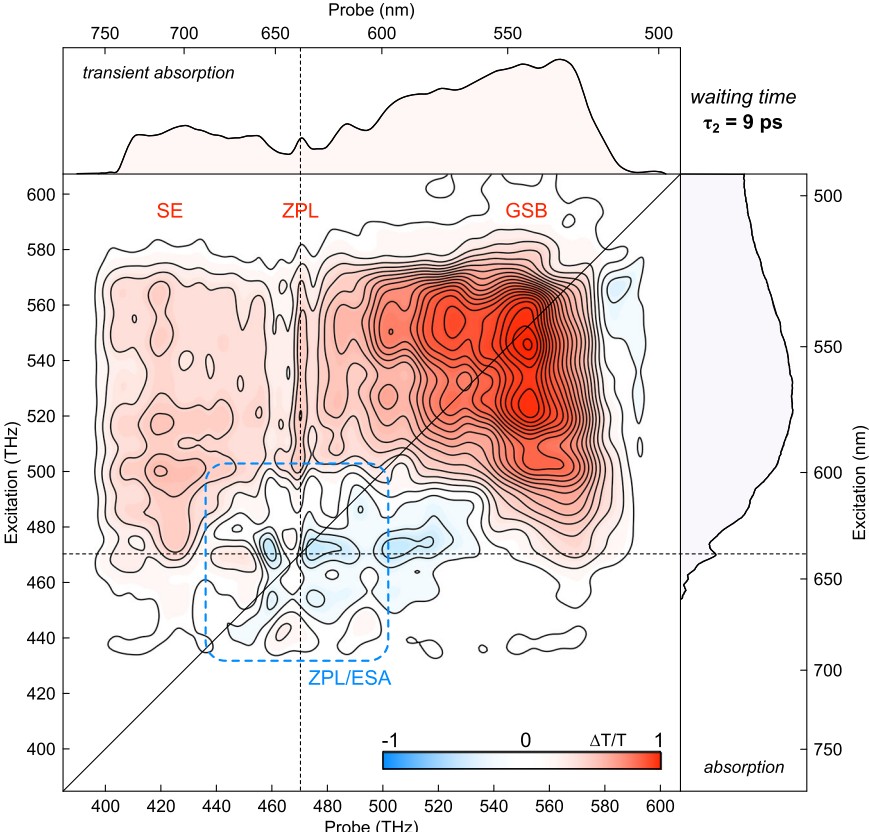

**Fig. 4 | Representative 2D electronic spectrum of the NV center at a waiting time ($\tau_2$) of 9 ps.** The four ESA peaks clustered around the ZPL excitation–ZPL emission cross peak are signatures of the LO phonon mediated dynamics of the JT distortion. Insets correspond to a TA projection obtained at the same waiting time (top) and the absorption spectrum of the NV center (right).

**Table 1 | A selection of peak separation energies from a vibronic analysis corresponding to qLVMs predicted by ab initio computations and previous spectral simulations**

| Energy (meV) | Frequency (THz) | Symmetry | Narrowband TA (meV)[34] | Computational[33] (G ≠ 0) (meV) | Computational (G = 0) (meV) |
|---|---|---|---|---|---|
| 36.3 | 9.4 | $e$ | – | 36.7 | 39.3 |
| 47.5 | 11.5 | $e$ | 47 | – | – |
| 66.2 | 16.0 | $a_1$ | 69 | 67.1 | – |
| 75.0 | 18.1 | $a_2$ | – | 78.1 | – |
| 85.3 | 20.6 | $e$ | 90 | – | 88.5 |

methods that can capture the intricate ultrafast dynamics of JT distortions may be one such technique.

The presence of the split-ESA peaks can only occur if the LO phonon extends the modulation of the JT distortion to neighboring NV centers as described in Fig. 1. The LO phonon couples neighboring NV centers together, and this interaction allows a sub-population of NV centers to be excited directly into an excited state with JT distorted geometry. As a consequence, the ESA peaks include the $a_1$ qLVM formally symmetry forbidden in direct absorption, and also lack the ultrafast dynamics observed in the stimulated emission signal (see: Supplementary Fig. 6). Based on the bandwidth of the LO phonon in the power spectrum of the TA (2 THz) and a diamond refractive index of 2.42, the coherence length of the LO phonon is 62 μm (3020 μm² area). The laser spot size used in these experiments was 150 μm and there are approximately $10^{17}$ NV centers per cubic cm at an average distance of a few nm, suggesting ~17% of the original spot size is available for subsequent absorption.

Comparing the results of the vibronic analysis to the power spectrum of the TA data results in an interesting discrepancy: the electronic energy levels of the NV center are strongly modulated by the LO phonon, but that modulation appears to be independent of the qLVMs. A consideration of the temperature dependence of NV center spectra alleviates this discrepancy. All of the spectra presented here were acquired at room temperature. Previous observations of the homogeneous broadening, PL polarization contrast, and excited-state relaxation properties of the ZPL at cryogenic temperatures have uncovered a $T^5$ dependence consistent with two-phonon Raman transitions between the adiabatic potential energy surfaces of the JT distortion[7]. Because every qLVM of the NV center has an energy larger than the JT stabilization energy, the population transfer through the conical intersection ensures that even at low temperature the PL polarization contrast is never 100%[10]. In contrast, measurements conducted specifically on the "lower branch" of the ZPL, i.e., the post-distortion potential energy surface, have returned $T^3$ dependence,

with a linear JT interaction and localized strain effects offered as potential explanations[7].

The ESA features in this report suggest an excited-state JT distortion that splits the geometry of the surrounding diamond lattice. In the absence of strain, excitations into the ZPL or higher-lying electronic states rapidly undergo nonradiative decay through the JT conical intersection into the symmetry-relaxed state. From there, the qLVMs and LO phonon have differing effects on the dynamics of the NV center. Two-phonon Raman transitions composed of qLVMs mediate the relaxation of the JT excited state, and ultimately the NV center spin-polarization cycle. They appear as static peak separations in the 2D electronic spectra, and we predict that the homogeneous broadening of the peaks in the 2D spectra should follow the same $T^5$ dependence at cryogenic temperatures. In contrast, the LO phonon disperses the local geometric distortion of the JT excited state throughout the diamond lattice. This promotes ESA signal from neighboring NV centers that absorb light directly into the symmetry-relaxed state. The origin of the ESA features are, essentially, a dynamically induced, static JT distortion, and based on the previous reports we expect the population decay of the ESA signal to have an approximately $T^3$ dependence, taking into account other minor sources of strain.

Recent reports of 2D ES measurements of NV and silicon vacancy (SiV) centers at cryogenic temperatures did not have the required spectral bandwidth to coherently excite LO phonons[35–37], however a similarly recent report on negatively charged SiV centers in diamond attributed the blue-shift of the ZPL to the expansion of the crystal lattice and found that these geometric effects had a $T^3$ dependence[38]. Interestingly, the optical transition to the lowest level of the JT-distorted electronic excited state of neutral SiV centers is forbidden[39]. It would be interesting to investigate in future work whether symmetry breaking through coherent LO phonons can activate this transition like the $a_1$ transition in the NV center. In general, the effects of LO-phonon modulation on the JT distortions of lattice and topological materials are under-reported, and the example of the NV center may indicate a previously untapped route for the exploration and design of these materials.

To probe the relationship between the LO phonon, the JT distortion, and the qLVMs of the NV center further, we conducted 2D ES across a series of waiting times that resulted in three-dimensional electronic spectroscopy (3D ES) data. These data yield time and frequency plots at specific optical transitions with minimal interference from other signal pathways. Supplementary Fig. 6 shows fitted decay traces of a 3D ES scan from 0 to 12 ps. Although a large non-resonant response obscures the TA and 2D ES at times earlier than 50 fs, the observation of progressively shorter rise times in the SE signal as the excitation energy approaches the ZPL suggests the same sub-50 fs vibronic dynamics observed in the prior report[12].

Focusing on the ZPL region in the 3D ES reveals significant retention of the inhomogeneous broadening of the LO-phonon modulated ESA features. Figure 5 highlights this dynamic ellipticity, where the ESA peak at 470 THz excitation and 480 THz emission for each 2D ES between 0 and 12 ps was fitted to a rotated Gaussian lineshape. The ellipticity decays from 1.6 to 0.2 in 1.55 ps, a spectral diffusion that is mediated by the LO phonon of the diamond. The decay time of the ESA peak—the LO-phonon dephasing time—is 3.83 ps. These dynamics, taken together, strongly resemble the 4.2 ps dynamic invoked as a vibronic shoulder in a previous 2D ES report[20], where the spectra were not phased and the waiting time ($\tau_2$) resolution was both more sparse and logarithmic.

One final observation in the 3D ES adds additional detail to the interaction of the JT distortion and the LO phonon. As seen in Fig. 6, the 1D power spectra of the 3D ES data contain several peaks. While these peaks may appear to be qLVM-like on first inspection, every peak is either the LO phonon or aliases arising from 3D ES scans with

temporal resolutions below the LO phonon period. Figure 6 includes data from many generations of our experimental setup, and a table of each spectrum's specific experimental parameters is given in the Supplemental Information. Strikingly, the 1D vibronic spectrum arising from the 3D ES data using 1 fs waiting time steps shows a series of peaks at harmonic frequencies higher than the LO phonon: overtones of the lattice vibration.

The overtone peaks are made all the more remarkable by their preferential modulation of the electronic transitions associated with the JT distortion. As seen in the magnitude and phase-resolved coherence spectra of the LO phonon fundamental (40.1 THz) and overtone (79.9 THz) frequencies, the ZPL and ESA peaks have greater modulation amplitudes in the second harmonic spectra than in the LO spectra. Coherence spectra depict the magnitude and phase of wavepacket motion, with nodes appearing at potential energy surface minima. The lack of nodes in the 40.1 THz coherence spectrum indicates that the LO phonon modulates the wavepacket globally, as suggested originally by the TA spectra. The overtone coherence spectrum has nodes specifically at the ZPL transition and its ESA counterparts, indicating that the LO phonon overtone couples specifically to the excited-state of the NV center and specifically to the JT distortion. In addition, the 2.2 THz splitting of the second overtone peak is similar to the 10 meV (2.5 THz) symmetry-relaxed potential energy well proposed by computation[33], and suggests that the overtone interacts with both coordinates of the JT distortion—analogous to a weak, low-frequency oscillation amplified by a strong, high-frequency signal in interference spectroscopy. The direct coupling of the LO phonon and the JT distortion has, to our knowledge, not been predicted by theory, described by computational work, or measured in existing steady-state optical experiments.

Taken together, the TA, 2D ES, and 3D ES data presented here suggest a highly dynamic energy relaxation landscape for NV centers in the first 5 ps after excitation. Combined LO phonon and overtone modulation distorts neighboring NV centers, allowing nominally optically forbidden transitions to become optically active and effectively coupling them to the initial population of excited NV centers. While the qLVMs of the JT distortion govern the spin-polarization of the system, the LO phonon modulates the excited state dynamics. These near-neighbor interactions elevate the LO phonon to an important coordinate of the JT distortion and should inform future theoretical work previously focused on isolated NV centers. Indeed, the existence of a JT distortion in a material system with strong phonon modulation is hardly unique to the NV center, and the dynamics reported here suggest a possible new avenue for the design of novel solid-state materials alongside the exploration of existing systems.

## Methods

Previous works detail the femtosecond four-wave mixing spectrometer and high-sensitivity detection methods used in these measurements[40,41]. The Supplementary Information details specific measurement parameters for each dataset presented in this report. For the NV center experiments, we incorporate a second amplification stage in the noncollinear optical parametric amplifier (NOPA) to increase the pulse energy to 88 nJ. Then two pairs of dispersion-compensating mirror pairs and a prism-based pulse shaper that incorporates a neural-network algorithm[42] compressed the pulse to 6.5 fs in duration as measured by transient-grating frequency resolved optical gating (TG-FROG). These technical upgrades were needed to obtain adequate signal-to-noise levels and to ensure the wavepacket dynamics were not contaminated with ground-state dynamics due to temporal dispersion.

In the transient-transmittance measurements used for 'phasing' of the 2D ES data, the maximum nonlinear signal, $\Delta T/T$, was 3%, and averaged over 2500 kinetic cycle pairs. We scanned the pump–probe interval from −0.5 to 12 ps in 1 fs steps. The acquired signal was free of

nonresonant response after about 50 fs. In the 2D ES measurements, we scanned $\tau_1$ from 0 to 90 fs in 1 fs steps. The phasing procedure yielded residuals that were 0.3% or better for most 2D electronic spectra.

The NV sample was generously obtained from Prof. Neil Manson's research group. A CVD single-crystal diamond blank was purchased from Element 6. Both samples were approximately 3 mm × 3 mm × 0.3 mm in dimension. Prior to each measurement, the NV center sample was

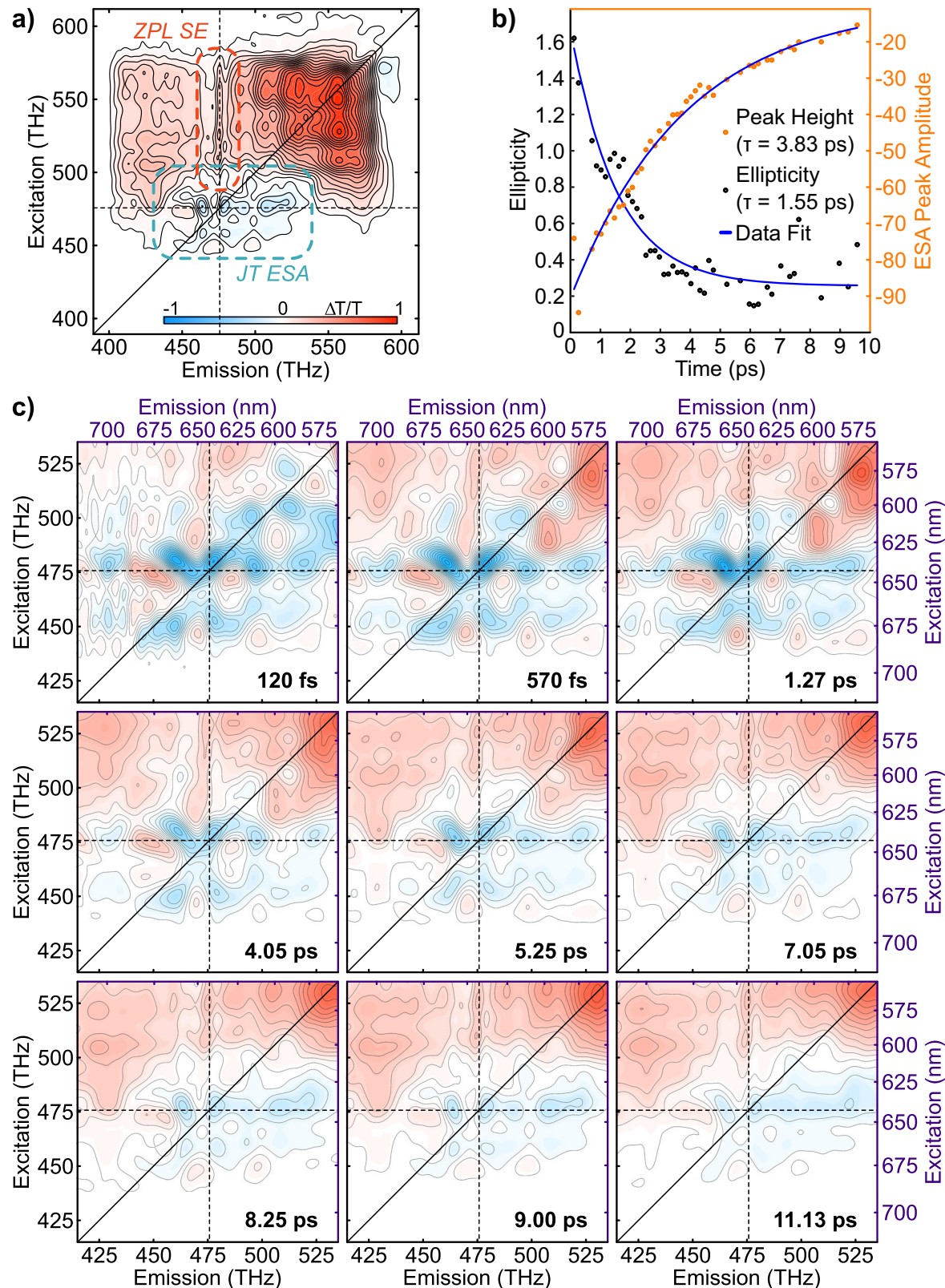

**Fig. 5 | A set of 2D electronic spectra showing the dynamics of the ZPL region.**
**a** Annotated 2D electronic spectrum of the NV center at a waiting time of 9 ps.
**b** Temporal traces of the ellipticity (black) and intensity (orange) of the ESA peak at 470 THz excitation and 480 THz emission. **c** Zoomed in presentation of spectral diffusion reducing the ellipticity of the ESA peaks near the ZPL region as the waiting time increases.

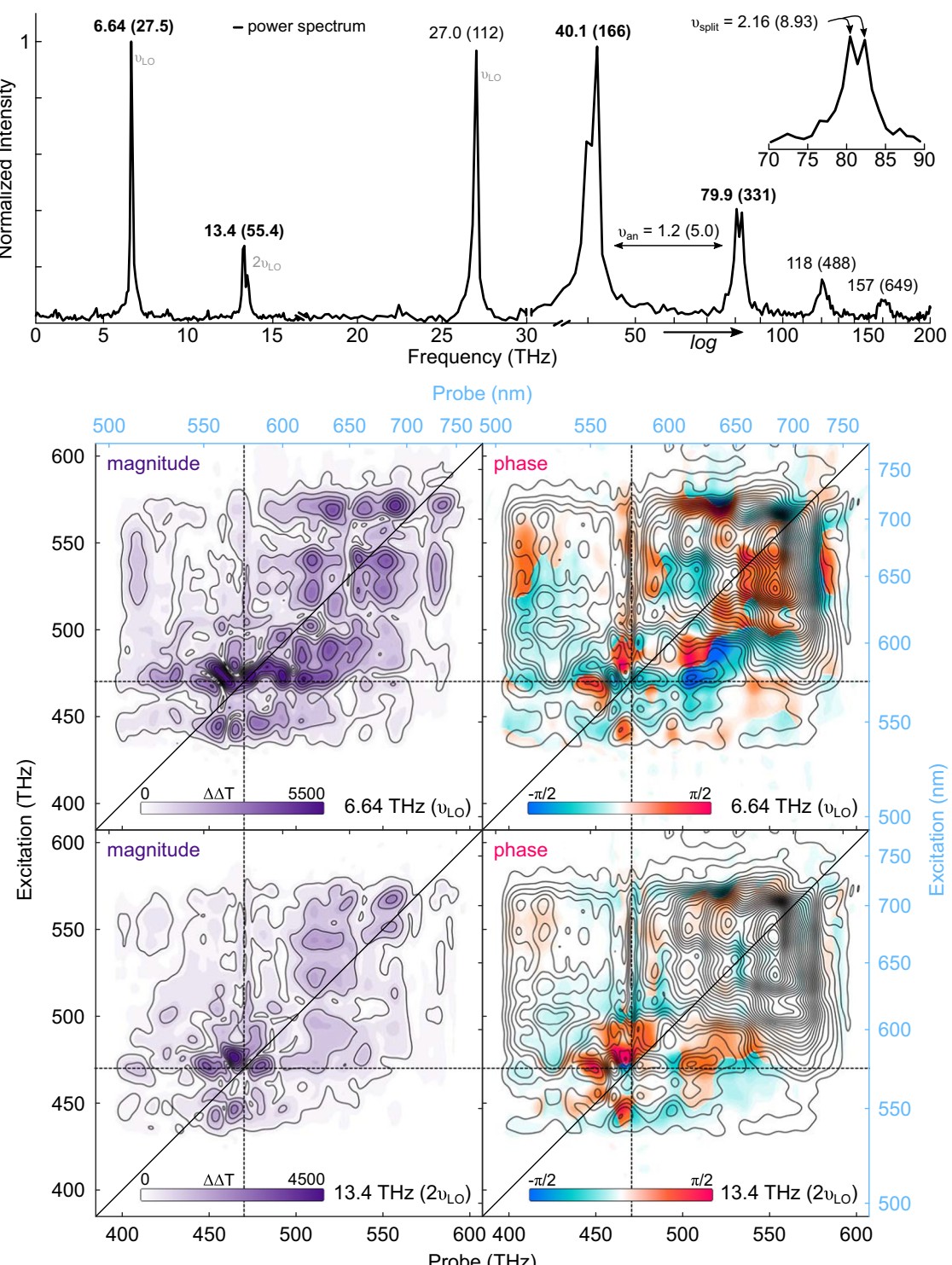

**Fig. 6 | Summary of the 3D ES data.** (top) Combined 1D power spectra of multiple 3D ES data showing the LO phonon and its overtones originating from two different measurements, normalized to their respective peak intensities. Peaks above 30 THz originate from a waiting time scan of 1 ps in 1 fs steps, while peaks below 30 THz originate from scans with >25 fs resolution. Peaks are labeled in THz (meV) and the identity of the fundamental peak is given in grey for the aliased peaks below 30 THz. (bottom) Phase-resolved coherence spectra showing the magnitude (purple) and phase (blue and orange) of the electronic transitions modulated by the LO phonon (40 THz, 6.64 THz aliased) and second harmonic (79.9 THz, 13.4 THz aliased).

washed in gently stirred, boiling piranha solution (3 $H_2SO_4$: 1 $H_2O_2$) for 12 h to reduce surface defects that cause significant scatter during the measurements. The Supplementary Information contains photographs of the NV center sample surface before and after cleaning and further details about the cleaning process. All spectroscopy measurements were conducted at ambient temperature of ~21 °C.

## Data availability

The four TA and 2D ES datasets used to generate the figures and conclusions for this report have been deposited in a Zenodo database under accession code doi:10.5281/zenodo.12740637. These data are available with minimal restrictions in accordance with the Creative Commons Attribution 4.0 International license. High-resolution

versions of the figures presented in this report are available on request by contacting William Carbery (wpcarbery@gmail.com).

## Code availability

The Matlab scripts used for the analysis of the time-resolved spectra is available in the same Zenodo repository (doi:10.5281/zenodo.12740637) that contains the data. A README file contains directory information.

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

## Acknowledgements

We thank Prof. Neil Manson for providing the NV sample and Gergő Thiering for discussions. The work at N.Y.U. was supported by the Alfred P. Sloan Foundation (DBT) and the National Science Foundation under CAREER Grant No. CHE–1552235 (DBT). R.U. acknowledges funding from the Max-Planck Society.

## Author contributions

R.U. proposed the study. D.B.T. supervised and obtained funding to support the project. W.P.C. and C.A.F. performed the measurements and data analysis. W.P.C. drafted the initial paper, and all authors contributed to the writing of the report.

## Funding

## Competing interests

The authors declare no competing interests.
