## [Peer Review File · Nature Communications]

The Phonon-Modulated Jahn–Teller Distortion of the Nitrogen Vacancy Centre in DiamondREVIEWER COMMENTS

Reviewer #1 (Remarks to the Author):

Review of “The Phonon-Modulated Jahn-Teller Distortion of the Nitrogen Vacancy Center in Diamond” by Carbery et al., manuscript ID NCOMMS-24-01731-T.

In this manuscript, Carbery et al. share interesting findings related to photoresponse of NV centers using a combination of transient absorption and 2D/3D electronic spectroscopy. The report is thorough and well-incorporates previous research findings. It appears that the work is very valuable in clearing up existing holes in our understanding of NV center photodynamics.

I provide comments below that may help the authors clarify and improve their manuscript further. I want to note that already the manuscript is very interesting and valuable for the community.

As far as I understand, the authors posit that the presence of nearby NV centers is that, upon optical excitation, these other centers get excited, and in addition to electronic excitation, they can also create LO phonons, which couple to the NV centers in a way which loosens the selection rules of absorption and emission.

1. I am wondering, for the sake of the veracity of this manuscript as well as the interest of the community, if the authors can comment on two important aspects of the phenomena described. This would be useful in corroborating the ideas presented in the manuscript as well as potentially provide ideas for the community going forward. First, as far as I can tell, authors perform all of their experiments at ambient temperatures. How do the authors expect these findings to depend on temperature?

2. Second, the authors' idea of nearby NV centers playing an important role is interesting. I'm wondering if there are any comments related to how the results might depend on the

density of NV centers in the system. On a related note, the authors comment that undoped diamond samples have LO signatures similar to those measured here, however are 4.5 times weaker in intensity. Is there some idea of what would change as a function of this LO phonon intensity, if one were to have a completely isolated NV center?

3. I'm interested in any underlying theoretical arguments for the results presented here. The authors write that at least some aspects of the findings were "unusual." It appears from what I can see that the experimental measurements performed are sound and well-studied. Nevertheless, can the authors comment on any theoretical aspects of the interaction of the LO phonon and the JT conical intersection? Is there a valuable theoretical insight that can help us rationalize why this "unusual" behavior is observed?

4. On a similar note, why in particular is the LO phonon so special? It was hard for me to appreciate this from reading the manuscript and perhaps a clear explanation will help to make the manuscript more accessible to a broad audience.

5. The idea that nearby NV centers impact an NV center of interest by being sources of LO phonons is interesting. Could the authors maybe comment, should this kind of behavior be expected for other defects in diamond? For example, group IV- and group IV0 emitters both have important Jahn-Teller effects which may be altered if this idea of LO phonon coupling affects their potential energy surfaces and selection rules. It's possible that in an experiment focused on these defects, NV centers are also getting excited which then may introduce these LO phonons. I imagine it's too challenging to say something even more broadly for Jahn-Teller defects in general.

Reviewer #2 (Remarks to the Author):

This manuscript reports on an ultrafast laser spectroscopy study of the NV center in diamond. The authors employ TA and CMDS experiments with unprecedented bandwidth.

The results are indeed spectacular. But the story is missing a lot and is not a compelling case for NComm.

1. Much is known about the spectroscopy and coherence aspects of NV in diamond. What key thing is missing that will be solved here or revealed here?

2. What is produced is a detailed dynamics picture of excited state relaxation. While this is a fine result, why is it that broadly interesting?

3. How do the CMDS experiments and in particular the 2D phase and amplitude maps explain about the nature of the phonons. And how does this explanation offer conclusive insight to a long standing problem.

The paper is very nice and should be published. It does not do well at convincing the reader of broad appeal.

Reviewer #3 (Remarks to the Author):

The properties of the negatively charged nitrogen (NV) vacancy in diamond attract the attention of researchers both from an applied and fundamental point of view. It is the latter that the reviewed work is devoted to. The authors very carefully established a number of basic NV properties using various techniques and for the first time discovered the effect of longitudinal optical phonons of the diamond lattice on the optical properties of the defect under study. My questions to the Authors are related to the interpretation of the results and the evaluation of the role of the Jahn-Teller effect. The Authors suggest that the vacancy complex in the excited state is subject to the Jahn-Teller effect, so that in this state there are three equivalent configurations with C_{3h} symmetry. Accordingly, there are many equivalent configurations and thus the results of observations should depend on the polarization of the exciting light. Has this dependence been observed? The second question concerns the magnitude of the splitting of electronic states under the action of a LO phonon. According to estimates from the References (23), the energy of the Jahn-Teller

stabilization is 25 meV, and the phonon energy is 40 meV. A change in the electronic state by such an amount can lead to the suppression of the Jahn-Teller effect and the restoration of the symmetry of the defect to C_{3v} . It is also unclear which type of effect - dynamic or static - is assumed by the Authors, they indicate the characteristic values of local phonons, but do not indicate the estimated value of the Jahn-Teller stabilization

Reviewer: 1

Comments:

In this manuscript, Carbery et al. share interesting findings related to photoresponse of NV centers using a combination of transient absorption and 2D/3D electronic spectroscopy. The report is thorough and well-incorporates previous research findings. It appears that the work is very valuable in clearing up existing holes in our understanding of NV center photodynamics.

I provide comments below that may help the authors clarify and improve their manuscript further. I want to note that already the manuscript is very interesting and valuable for the community.

As far as I understand, the authors posit that the presence of nearby NV centers is that, upon optical excitation, these other centers get excited, and in addition to electronic excitation, they can also create LO phonons, which couple to the NV centers in a way which loosens the selection rules of absorption and emission.

1. I am wondering, for the sake of the veracity of this manuscript as well as the interest of the community, if the authors can comment on two important aspects of the phenomena described. This would be useful in corroborating the ideas presented in the manuscript as well as potentially provide ideas for the community going forward. First, as far as I can tell, authors perform all of their experiments at ambient temperatures. How do the authors expect these findings to depend on temperature?

Here the reviewer expresses curiosity about the temperature and how that would affect the results. Indeed, all the measurements detailed in the manuscript were conducted at room temperature. Upon considering Reviewer 1's comments, we realized that a discussion of the temperature dependence provided a far superior conclusion than the one we had initially offered. To that end, three paragraphs on pages 15 and 16 discuss the temperature dependence in reference to the Doherty review published in 2013 (which we think provides an excellent review to scientists with a broad level of knowledge) and work referenced therein. These paragraphs essentially replace the previous two concluding paragraphs.

To summarize the findings presented in this report, it is instructive to consider the temperature dependence of the NV center properties. All of the spectra presented here were acquired at room temperature. Previous observations of the homogeneous broadening, PL polarization contrast, and excited-state relaxation properties of the ZPL at cryogenic temperatures have uncovered a T^5 dependence consistent with two-phonon Raman transitions between the adiabatic potential energy surfaces that constitute the JT distortion.⁷ Because every qLVM of the NV center has an energy larger than the JT stabilization energy, the population transfer through the conical intersection ensures that even at low temperature the PL polarization contrast is never 100%.¹⁰ In contrast, measurements conducted specifically on the "lower branch" of the ZPL, i.e., the post-distortion potential energy surface, have returned T^8 dependence, with a linear JT interaction and localized strain effects offered as potential explanations.⁷

The ESA features in this report suggest an excited-state JT distortion that splits the geometry of the surrounding diamond lattice. In the absence of strain, excitations into the ZPL or higher-lying electronic states rapidly undergo nonradiative decay through the JT conical intersection into the symmetry-relaxed state. From there, the qLVMs and LO phonon have differing effects on the

dynamics of the NV center. Two-phonon Raman transitions composed of qLVMS mediate the relaxation of the JT excited state, and ultimately the NV center spin-polarization cycle. They appear as static peak separations in the 2D electronic spectra, and we predict that the homogeneous broadening of the peaks in the 2D spectra should follow the same T^5 dependence at cryogenic temperatures.

By contrast, the LO phonon disperses the local geometric distortion of the JT excited state throughout the diamond lattice. This promotes ESA signal from neighboring NV centers that absorb light directly into the symmetry-relaxed state. The vibronic and coherence spectra indicate that, at room temperature, the LO-phonon mediated dynamics dominate both the excited-state relaxation dynamics of the JT distortion and the geometry of the NV center and diamond lattice. The origin of the ESA features are, essentially, a dynamically induced, static JT distortion, and based on the previous reports we expect the population decay of the ESA signal to have an approximately T^8 dependence, taking into account other minor sources of strain. Recent reports of 2D ES measurements of NV and silicon vacancy (SiV) centers at cryogenic temperatures did not have the required spectral bandwidth to coherently excite LO phonons,³³⁻³⁵ however a similarly recent report on negatively charged SiV centers in diamond attributed the blue-shift of the ZPL to the expansion of the crystal lattice and found that these geometric effects had a T^8 dependence.³⁴ Interestingly, the optical transition to the lowest level of the JT-distorted electronic excited state of neutral SiV centers is forbidden.³⁷ It would be interesting to investigate in future work, whether symmetry breaking through coherent LO phonons can activate this transition.

Taken together, the data presented here suggest a highly dynamic energy relaxation landscape for NV centers in the first 5 ps after excitation. The combined LO phonon and overtone modulation distorts neighboring NV centers, allowing nominally optically forbidden transitions to become optically active and effectively coupling them to the initial population of excited NV centers. These near-neighbor interactions elevate the LO phonon to an important coordinate of the JT distortion, and can inform theoretical work previously focused on isolated NV centers. The transmission of local distortions from a dynamic JT across a crystal lattice also has implications for material defects beyond the NV center. Systems with a similarly energetic JT distortion and a crystal lattice and material defect concentration able to support coupling by LO phonons should display similar transmission of the JT dynamics across the geometric space of the crystal, potentially opening up new use cases for NV centers and related systems.

In the updated Conclusion section, we make two predictions that we hope can be evaluated by other researchers:

- 1) the homogenous broadening of the 2D ES peaks specifically assigned to qLVMS should have a T^5 dependence in accordance with two-phonon Raman transitions between the excited state energy levels around the JT distortion, and
- 2) the population decay of the excited-state absorption signals should have a T^8 dependence in accordance with a linear JT effect and other crystal strain properties proposed by others in the NV center and silicon centers.

One of our key findings is that coherently excited LO phonons make possible observing the nominally optically forbidden vibronic transition to a_1 in 3E from 3A_2 . Due to the very large LO

phonon energy of 165 meV, there is practically no change in its linewidth when going to cryogenic temperatures, see Phys. Rev. B 4, 2493 (1971), hence the dephasing time will not change either. We thus do not expect the observed dynamics of the LO phonon to have significant temperature dependence. Similarly, the appearance of the ESA features and their modulation should remain consistent at cryogenic temperature as these properties are determined by coherent LO phonons. There could be a decrease in linewidth of the vibronic transitions at lower temperatures, although generally this effect is much less pronounced as compared to pure electronic transitions (i.e. ZPLs).

For the sake of brevity in the Conclusion section—and because it is briefly mentioned in the discussion of the 2D ES—we did not add that unphased spectra of photon-echo or other pump-probe measurements, including four-wave mixing experiments, could lead to spurious temperature dependence due to the ESA features. Those features are nearly symmetric around the ZPL and in magnitude spectra could have significant broadening effects on the lineshape, while having a different underlying temperature dependence. We again thank the reviewer for this helpful comment that clarified much of our initial consideration.

2. Second, the authors' idea of nearby NV centers playing an important role is interesting. I'm wondering if there are any comments related to how the results might depend on the density of NV centers in the system. On a related note, the authors comment that undoped diamond samples have LO signatures similar to those measured here, however are 4.5 times weaker in intensity. Is there some idea of what would change as a function of this LO phonon intensity, if one were to have a completely isolated NV center?

Here the reviewer poses another interesting question, which is the anticipated dependence on NV density. On a very basic level, the overall signal strength would decrease with decreasing defect density. For a truly isolated NV center we would still expect to see the LO phonon in the TA and 3D ES data, but perhaps not the ESA signal (and therefore none of the interesting coherence spectra!).

3. I'm interested in any underlying theoretical arguments for the results presented here. The authors write that at least some aspects of the findings were "unusual." It appears from what I can see that the experimental measurements performed are sound and well-studied. Nevertheless, can the authors comment on any theoretical aspects of the interaction of the LO phonon and the JT conical intersection? Is there a valuable theoretical insight that can help us rationalize why this "unusual" behavior is observed?

We briefly describe some of the NV center's anomalous optical properties in the Introduction, but in general the ultrafast spectroscopic data for NV centers remain limited. In terms of theory—the reviewer brings up an interesting point—we are not aware of any theoretical work that explicitly models the effects of near-neighbor interactions on JT distortions. The present contribution will hopefully inspire and motivate theoretical work on those considerations. On pages 4–5 we have added some detail on the LO phonon (as discussed in reviewer 1, comment 4):

We find evidence of the computationally predicted qLVMS in the peak separations of individual 2D ES spectra but determine through coherence spectra that only the LO phonon and

its overtones directly modulate the JT distortion. A schematic representation of the NV center's relevant electronic states is shown in Figure 1, alongside the adiabatic potential energy surfaces proposed for the ESA features observed in our spectra. Our findings suggest that the longitudinal optical (LO) phonon of the diamond lattice dominates the dynamics of the JT distortion at room temperature, bypassing the qLVMs and coupling neighboring NV centers through the diamond lattice.

and some of the theoretical considerations are mentioned in the new Conclusion along with the response to reviewer comment 5.

4. On a similar note, why in particular is the LO phonon so special? It was hard for me to appreciate this from reading the manuscript and perhaps a clear explanation will help to make the manuscript more accessible to a broad audience.

For the spin properties of the NV center—which has been the primary focus in the literature—the LO phonon may have a limited role, especially at cryogenic temperatures. It is interesting here because the LO phonon is so rarely considered in theoretical and computational treatments of the NV center, and yet the diamond lattice is what makes the JT distortion in the NV center so interesting. Molecular JT distortions do not have near-neighbor effects or phonon modes! On page 4 we have added some supporting language to emphasize the LO phonon's role in our data, although we stop short of a full discussion for the sake of brevity. We hope that, along with a better conclusion, this is sufficient to emphasize the LO phonon's role in the presented data.

5. The idea that nearby NV centers impact an NV center of interest by being sources of LO phonons is interesting. Could the authors maybe comment, should this kind of behavior be expected for other defects in diamond? For example, group IV- and group IV0 emitters both have important Jahn-Teller effects which may be altered if this idea of LO phonon coupling affects their potential energy surfaces and selection rules. It's possible that in an experiment focused on these defects, NV centers are also getting excited which then may introduce these LO phonons. I imagine it's too challenging to say something even more broadly for Jahn-Teller defects in general.

Reviewer 1 once again poses an excellent question. We were hesitant to extrapolate too much from the NV center, as that system is complicated enough for analysis. On page 17 in the revised Conclusion section, we mention that the NV center results have implications for other defects with dynamic Jahn-Teller distortions and concentrations large enough to support near-neighbor effects.

Recent reports of 2D ES measurements of NV and silicon vacancy (SiV) centers at cryogenic temperatures did not have the required spectral bandwidth to coherently excite LO phonons,³⁸⁻³⁵ however a similarly recent report on negatively charged SiV centers in diamond attributed the blue-shift of the ZPL to the expansion of the crystal lattice and found that these geometric effects had a T^8 dependence.³⁶ Interestingly, the optical transition to the lowest level of the JT-distorted electronic excited state of neutral SiV centers is forbidden.³⁷ It would be interesting to investigate in future work, whether symmetry breaking through coherent LO phonons can activate this transition.

Reviewer: 2

Comments:

This manuscript reports on an ultrafast laser spectroscopy study of the NV center in diamond. The authors employ TA and CMDS experiments with unprecedented bandwidth. The results are indeed spectacular. But the story is missing a lot and is not a compelling case for NComm.

1. Much is known about the spectroscopy and coherence aspects of NV in diamond. What key thing is missing that will be solved here or revealed here?

We agree with Reviewer 2 that the NV center dynamics are well established. The consensus (correct, in our view) is that the JT distortion of the NV center is in the excited state and is dynamic—however the majority of this description is founded on measurements of the ZPL linewidth and relaxation decay—both of which are potentially susceptible to contamination by the ESA features found in our work. We have added a sentence on page 9 that makes that point and appreciate the Reviewer’s comment that the story was not being optimally presented.

Phasing the 2D ES is particularly important for the NV center because the ESA peaks are symmetric across the ZPL and therefore *disappear in magnitude spectra*. **Measurements of the ZPL linewidth and relaxation decay provide the foundation for discussing NV center dynamics, and the ESA features have the potential to disrupt interpretations of both measurements if not properly phased and analyzed.**

In addition, we have updated the Conclusion section (pages 15 and 16) that tie in our work better with the existing literature.

As mentioned in response to Reviewer 1, one of our key findings is that coherently excited LO phonons make possible observing the nominally optically forbidden vibronic transition to the a_1 vibronic level (i.e. tunneling splitting) of the 3E manifold, which we find to be 36 meV above the 3E vibrational ground state, in line with previous calculations. The JT stabilization energy has also not yet been determined experimentally but there are two DFT simulation papers that reported calculations. While we currently don’t see a way to draw conclusions regarding that from our data, we hope that future work can address this. On page 11 we have added:

Two of the ESA peaks are red-shifted from the ZPL by 8.8 THz, a frequency that *ab initio* computations²² predict to be the lowest-energy qLVM a_1 , i.e. the JT tunneling splitting. **Although the transition to the a_1 qLVM is forbidden in the initial excitation, symmetry breaking by the JT distortion**

allows the transition. This suggests that the ESA peaks arise from ZPL transitions between the ground (6A_1) and excited state (8E) after the JT distortion displaces the local geometry of the NV center. Depending on the simulation, the JT stabilization energy is below (25 meV)³² or slightly above (42 meV)⁵ the a₁ qLVM, and it would be interesting if future theoretical work could use the observed ESA features to predict optical signatures in 2D ES or related methods that would distinguish between these two scenarios. Nonadiabatic wavepacket propagation methods that can capture the intricate ultrafast dynamics of JT distortions may be one such technique.

2. What is produced is a detailed dynamics picture of excited state relaxation. While this is a fine result, why is it that broadly interesting?

The dynamics presented in our report give additional depth to the LO phonon interaction in the NV center and emphasize that this effect is not captured in existing computational and theoretical work, for understandable reasons given the computational restraints a decade ago. We now mention this directly on page 4 before the results are presented.

We find evidence of the computationally predicted qLVMs in the peak separations of individual 2D ES spectra but determine through coherence spectra that only the LO phonon and its overtones directly modulate the JT distortion.

The rewritten Conclusion also ends on page 17 with the extrapolation of the LO phonon effect on other material defects that may also have interconnected JT distortions.

Recent reports of 2D ES measurements of NV and silicon vacancy (SiV) centers at cryogenic temperatures did not have the required spectral bandwidth to coherently excite LO phonons,³⁸⁻³⁵ however a similarly recent report on negatively charged SiV centers in diamond attributed the blue-shift of the ZPL to the expansion of the crystal lattice and found that these geometric effects had a T⁸ dependence.³⁴ Interestingly, the optical transition to the lowest level of the JT-distorted electronic excited state of neutral SiV centers is forbidden.³⁷ It would be interesting to investigate in future work, whether symmetry breaking through coherent LO phonons can activate this transition.

3. How do the CMDS experiments and in particular the 2D phase and amplitude maps explain about the nature of the phonons. And how does this explanation offer conclusive insight to a long standing problem.

The paper is very nice and should be published. It does not do well at convincing the reader of broad appeal.

We appreciate the Reviewer's comments on the accessibility of our report. The phase maps in particular offer good evidence that the LO phonon is *directly* coupled to the NV center's JT distortion, which is not something that - to our knowledge - has ever been predicted by theory,

computation, or steady-state optical experiments. We have added a small section before the conclusion on page 14 to further describe the phase maps and CMDS experiments.

The overtone coherence spectrum has nodes specifically at the ZPL transition and its ESA counterparts, indicating that the LO phonon overtone couples specifically to the excited state of the NV center and specifically to the JT distortion. In addition, the 2.2 THz splitting of the second overtone peak is similar to the 10 meV (2.5 THz) symmetry-relaxed potential energy well proposed by computation,²² and suggests that the overtone interacts with both coordinates of the JT distortion — analogous to a weak, low-frequency oscillation amplified by a strong, high-frequency signal in interference spectroscopy. The direct coupling of the LO phonon and the JT distortion has, to our knowledge, not been predicted by theory, described by computational work, or measured in existing steady-state optical experiments.

The ultrabroadband nature of the spectra presented here should increase the appeal of the report to a broader audience, as phase maps with this spectral and frequency resolution remain rare.

Reviewer: 3

Comments:

The properties of the negatively charged nitrogen (NV) vacancy in diamond attract the attention of researchers both from an applied and fundamental point of view. It is the latter that the reviewed work is devoted to. The authors very carefully established a number of basic NV properties using various techniques and for the first time discovered the effect of longitudinal optical phonons of the diamond lattice on the optical properties of the defect under study. My questions to the Authors are related to the interpretation of the results and the evaluation of the role of the Jahn-Teller effect. The Authors suggest that the vacancy complex in the excited state is subject to the Jahn-Teller effect, so that in this state there are three equivalent configurations with C_{1h} symmetry. Accordingly, there are many equivalent configurations and thus the results of observations should depend on the polarization of the exciting light. Has this dependence been observed?

Reviewer 3 poses two good questions. The first is in regard to Jahn-Teller effect and the configurations with C_{1h} symmetry and whether other researchers have observed differences using polarized light. We have made efforts in the updated Conclusion section on Pages 15 and 16 to specifically mention “symmetry relaxed state” when talking about the JT distortion and the energy relaxation pathway because we were clear previously when we discussed the JT distortion. In terms of the polarization effects in the NV center, the luminescence resulting from excitation with either linear polarization is always converted with approximately 40% efficiency into the other polarization state, even at cryogenic temperatures. This is well established by Fu et. al. (Ref. 9 in our report) and elaborated on in the Doherty review (Ref. 6 in our report). Our laser was linearly polarized on all four beams and the NV sample was inserted at a random orientation, so we expect to see little polarization dependence on our results. At room temperature, the polarization is almost entirely scrambled because the NV center interconverts across all of the configurations of the JT distortion.

The second question concerns the magnitude of the splitting of electronic states under the action of a LO phonon. According to estimates from the References (23), the energy of the Jahn-Teller stabilization is 25 meV, and the phonon energy is 40 meV. A change in the electronic state by such an amount can lead to the suppression of the Jahn-Teller effect and the restoration of the symmetry of the defect to C_{3v} . It is also unclear which type of effect - dynamic or static - is assumed by the Authors, they indicate the characteristic values of local phonons, but do not indicate the estimated value of the Jahn-Teller stabilization.

Here the reviewer is curious about whether the Jahn-Teller effect is dynamic or static and about the stabilization energy. We have adjusted statements on pages 3,

The dynamic Jahn-Teller distortion also explains many of the NV center’s anomalous optical properties.

and on pages 15 and 16 to explicitly mention that the JT distortion is dynamic, i.e. that it is an excited-state distortion and is not isolatable in any of the potential energy surface minima of the resulting JT state geometry. The reviewer is correct that the phonon energy can return the symmetry of the JT distortion, and this is generally the consensus view that population transfer between the JT distortion minima can occur through the JT conical intersection, and that this interaction is largely what

accounts for the polarization scrambling even at low temperature. We have also included a clarification on page 11 that states the stabilization energy of the JT distortion is predicted to be 25 meV (6.6 THz) or 42 meV (10.2 THz):

Two of the ESA peaks are red-shifted from the ZPL by 8.8 THz, a frequency that *ab initio* computations²² predict to be the lowest-energy qLVM a₁, i.e. the JT tunneling splitting. Although the transition to the a₁ qLVM is forbidden in the initial excitation, symmetry breaking by the JT distortion allows the transition. This suggests that the ESA peaks arise from ZPL transitions between the ground (³A₂) and excited state (³E) *after the JT distortion displaces the local geometry of the NV center*. Depending on the simulation, the JT stabilization energy is below (25 meV)²² or slightly above (42 meV)⁵ this qLVM, and it would be interesting if future theoretical work could use the observed ESA features to predict optical signatures in 2D ES or related methods that would distinguish between these two scenarios. Nonadiabatic wavepacket propagation methods that can capture the intricate ultrafast dynamics of JT distortions may be one such technique.

REVIEWERS' COMMENTS

Reviewer #1 (Remarks to the Author):

Review #2 of “The Phonon-Modulated Jahn-Teller Distortion of the Nitrogen Vacancy Center in Diamond” by Carbery et al., manuscript ID NCOMMS-24-01731-T.

I appreciate the authors' effort and consideration of my questions and comments. I am generally satisfied with their responses. Below are some additional considerations that might improve the manuscript further.

1. I appreciate the authors' response about the temperature dependence. I've been considering the updated manuscript and my concern is that the current format, in particular of adding in the temperature dependence in the very beginning of the conclusion section, might lead to confusion. Readers may look to the conclusion for a brief overview of the work and a take-home message. However the authors currently instead lead with a discussion of temperature, which is a foreign topic up to this point in the paper. I wonder if the authors' might be better off putting this discussion in a previous section or arranging things differently?

2. I appreciate the comment from another reviewer about readability and appealing to a broad audience, and I also appreciate the authors' effort to do so. I also would appreciate this aim. I think that ideas discussed in the correspondence with my previous report as well as some of the ideas present in the current version of the manuscript could help make the research more generally interesting. Specifically, there is the distinction between Jahn-Teller in molecules and Jahn-Teller in crystals. The authors make this distinction in the opening paragraphs of their work, i.e., the lattice introduces bulk phonon modes. As the authors find, the LO phonon mode in particular can couple to and apparently is important to understanding transitions/dynamics involving Jahn-Teller distortions. As the authors point out, the inclusion of phonon modes arising from the lattice and how they affect Jahn-Teller-related transitions has not been well-appreciated either in experiment or theory previously. This kind of phenomena, albeit maybe to differing extents, could be present in a

wide variety of solid-state defects where JT distortions could play a role, for example not only defects in diamond but also those in hBN, SiC or TMDCs. I would just encourage the authors to keep this big picture in mind if possible, as much of the text and figures can be seen as very technical otherwise. I do believe this big picture perspective is already improved from the first submission of the manuscript, though.

Reviewer #1 (Remarks on code availability):

I cannot review the code because it is currently in an "embargoed" state.

Reviewer #2 (Remarks to the Author):

This experiment is indeed remarkable and it is pleasing to see in the revisions how the manuscript has evolved to better highlight the quality and significance of the work. I agree with the author that a large amount of NV work is based upon extremely simple spectroscopy which can only reveal an extremely simple picture. The authors have nicely employed state of the art in spectroscopy to reveal new physics that is broadly interesting as presented.

Reviewer #3 (Remarks to the Author):

I have read the Authors' responses to my comments. In my opinion, the Authors' responses have made the text of the article more understandable for a wide range of researchers. The article can be accepted for publication.

REVIEWERS' COMMENTS

Reviewer #1 (Remarks to the Author):

Review #2 of “The Phonon-Modulated Jahn-Teller Distortion of the Nitrogen Vacancy Center in Diamond” by Carbery et al., manuscript ID NCOMMS-24-01731-T.

I appreciate the authors' effort and consideration of my questions and comments. I am generally satisfied with their responses. Below are some additional considerations that might improve the manuscript further.

1. I appreciate the authors' response about the temperature dependence. I've been considering the updated manuscript and my concern is that the current format, in particular of adding in the temperature dependence in the very beginning of the conclusion section, might lead to confusion. Readers may look to the conclusion for a brief overview of the work and a take-home message. However the authors currently instead lead with a discussion of temperature, which is a foreign topic up to this point in the paper. I wonder if the authors' might be better off putting this discussion in a previous section or arranging things differently?

We agree with Reviewer 1 that, when rereading the conclusion, the discussion of the temperature dependence is abrupt. We have moved this discussion to the end of the 2D ES section of the paper, starting at the bottom of page 11. The first two sentences are:

Comparing the results of the vibronic analysis to the power spectrum of the TA data results in an interesting discrepancy: the electronic energy levels of the NV center are strongly modulated by the LO phonon, but that modulation appears to be independent of the qLVMs. A consideration of the temperature dependence of NV center spectra alleviates this discrepancy.

2. I appreciate the comment from another reviewer about readability and appealing to a broad audience, and I also appreciate the authors' effort to do so. I also would appreciate this aim. I think that ideas discussed in the correspondence with my previous report as well as some of the ideas present in the current version of the manuscript could help make the research more generally interesting. Specifically, there is the distinction between Jahn-Teller in molecules and Jahn-Teller in crystals. The authors make this distinction in the opening paragraphs of their work, i.e., the lattice introduces bulk phonon modes. As the authors find, the LO phonon mode in particular can couple to and apparently is important to understanding transitions/dynamics involving Jahn-Teller distortions. As the authors point out, the inclusion of phonon modes arising from the lattice and how they affect Jahn-Teller-related transitions has not been well-appreciated either in experiment or theory previously. This kind of phenomena, albeit maybe to differing extents, could be present in a wide variety of solid-state defects where JT distortions could play a role, for example not only defects in diamond but also those in hBN, SiC or TMDCs. I would just encourage the authors to keep this big picture in mind if possible, as much of the text and figures can be seen as very technical otherwise. I do believe this big picture perspective is already improved from the first submission of the manuscript, though.

Without adding too much length or interrupting the flow of the report, we have modified a few paragraphs to reemphasize the novelty of JT distortions in crystals and the relative paucity of computational or experimental studies on them.

On Page 4: The NV center is a lattice system not a molecular system, and JT distortions trapped in solid-state materials are less researched relative to their molecular counterparts. [...] While the results of this report are unique to the NV center, the phonon modulation of JT distortions likely occurs in other lattice, solid-state, and topological materials.

On Page 13: It would be interesting to investigate in future work whether symmetry breaking through coherent LO phonons can activate this transition like they do for the a_1 transition in the NV center. In general, the effects of LO-phonon modulation on the JT distortions of lattice and topological materials are under-reported, and the example of the NV center may indicate a previously untapped route for the exploration and design of these materials.

On Page 17: While the qLVMS of the JT distortion govern the spin-polarization of the system, the LO phonon modulates the excited state dynamics. These near-neighbor interactions elevate the LO phonon to an important coordinate of the JT distortion and should inform future theoretical work previously focused on isolated NV centers. Indeed, the existence of a JT distortion in a material system with strong phonon modulation is hardly unique to the NV center, and the dynamics reported here suggest a possible new avenue for the design of novel materials and the exploration of existing systems.

These additions should remind the reader that the NV center is unique to our data but not unique as a system, and that some of the dynamics reported here are likely applicable in common material systems with extensive research backgrounds. We again thank the reviewer for the considered recommendations!

Reviewer #1 (Remarks on code availability):

I cannot review the code because it is currently in an "embargoed" state.

The code was embargoed only until the end of August as a precaution before publication. It is currently available as of Sep 1st, 2024 on Zenodo.

Reviewer #2 (Remarks to the Author):

This experiment is indeed remarkable and it is pleasing to see in the revisions how the manuscript has evolved to better highlight the quality and significance of the work. I agree with the author that a large amount of NV work is based upon extremely simple spectroscopy which can only reveal an extremely simple picture. The authors have nicely employed state of the art in spectroscopy to reveal new physics that is broadly interesting as presented.

We appreciate Reviewer 2's previous recommendations and are happy that the revised manuscript better conveys our work to a broader scientific audience.

Reviewer #3 (Remarks to the Author):

I have read the Authors' responses to my comments. In my opinion, the Authors' responses have made the text of the article more understandable for a wide range of researchers. The article can be accepted for publication.

We appreciate Reviewer 3's previous recommendations and the time taken to review the manuscript.